# FITNET’s Internet-Based Cognitive Behavioural Therapy Is Ineffective and May Impede Natural Recovery in Adolescents with Myalgic Encephalomyelitis/Chronic Fatigue Syndrome. A Review

**DOI:** 10.3390/bs7030052

**Published:** 2017-08-11

**Authors:** Simin Ghatineh, Mark Vink

**Affiliations:** 1Biochemist, London TW11, UK; siminghatinehdr@gmail.com; 2Family Physician, Soerabaja Research Center, 1096 HH Amsterdam, The Netherlands

**Keywords:** chronic fatigue syndrome, FITNET Trial, cognitive behaviour therapy, graded exercise therapy, internet-based cognitive behavioural therapy, myalgic encephalomyelitis, recovery

## Abstract

The Dutch Fatigue In Teenagers on the interNET (FITNET) study claimed that after 6 months, internet based cognitive behaviour therapy in adolescents with Myalgic Encephalomyelitis/Chronic Fatigue Syndrome (ME/CFS), led to a 63% recovery rate compared to 8% after usual care, and that this was maintained at long term follow up (LTFU). Our reanalysis shows that their post-hoc definition of recovery included the severely ill, the unblinded trial had no adequate control group and it used lax selection criteria as well as outcomes assessed via questionnaires rather than objective outcomes, further contributing to exaggerated recovery figures. Their decision not to publish the actometer results might suggest that these did not back their recovery claims. Despite these bias creating methodological faults, the trial still found no significant difference in recovery rates (“~60%”) at LTFU, the trial’s primary goal. This is similar to or worse than the documented 54–94% spontaneous recovery rates within 3–4 years, suggesting that both FITNET and usual care (consisting of cognitive behaviour and graded exercise therapies) are ineffective and might even impede natural recovery in adolescents with ME/CFS. This has implications for the upcoming costly NHS FITNET trial which is a blueprint of the Dutch study, exposing it to similar biases.

## 1. Introduction

Myalgic Encephalomyelitis (ME) also known as Chronic Fatigue Syndrome (CFS) or ME/CFS is a debilitating multisystem disease as concluded by the American Institute of Medicine (IOM) in February 2015 after an extensive review of the literature [1]. It is characterized by rapid (muscle) fatigability after trivial exertion with an abnormally delayed recovery, pain, neurological complaints and autonomic and immunological dysfunction [2], often resulting in severe functional limitations [3] rendering at least 25% of patients house and/or bed bound [1]. The IOM also concluded that there is no effective treatment for it [1] and the American Federal Agency for Healthcare Research and Quality, removed its recommendation for CBT and GET in July 2016, after concluding that there is no evidence that these treatments are effective [4]. Patient surveys from ME associations over the last few decades including three recent ones from the British, Norwegian and Dutch involving more than 3000 patients [5,6,7], and also a survey of young ME patients [8], have shown that there is no effective treatment/cure for this disease. All this is in contrast with the conclusion of the Dutch FITNET trial that 60% of participants recovered after treatment with internet-based CBT [9]. This has led to the NHS allocating almost £1 million of funding, to replicate the Dutch study in the UK [10]. 

In this article we re-analysed the FITNET trial, in an attempt to find out whether internet based CBT can lead to actual recovery from CFS in children/adolescents and whether there were any other important findings and if so, what they were. 

* The FITNET protocol used both terms (ME and CFS) [11] but as the subsequent publications only used the term CFS [9,12], we will do the same in this analysis to avoid any confusion. 

## 2. Summary of the FITNET Trial

The FITNET (Fatigue In Teenagers on the interNET) trial was registered as a randomised controlled trial involving 135 adolescents with CFS aged 12–18 years at the trial’s start. It compared the effectiveness of internet-based CBT with that of usual care, as although face-to-face CBT is reported to be effective in two-thirds of adolescents with CFS, it requires specialized therapeutic skills that are not always available within driving distance [9]. The trial’s primary goal was the recovery rate at long-term follow up (LTFU). Primary outcomes were fatigue severity, physical functioning and school attendance assessed via questionnaires. The trial concluded that after six months, i.e., at end of treatment, 60% of participants had recovered in the Internet based CBT group and only 8% in the usual care group. At LTFU (mean of 2.7 years after treatment’s start) the positive effects of FITNET were maintained, 58.9% of adolescents had recovered from CFS. Usual care led to similar recovery rates, although these rates were achieved at a slower pace [9].

## 3. Issues with Trial Design 

### 3.1. The Protocol 

The trial was registered with the clinical trials registry on 29/10/2007 as a randomised controlled trial [13]. Participants were recruited between 2008 and 2010 [12], but the trial’s protocol was not submitted until 2011 [11] even though “A fundamental principle in the design of randomized trials involves setting out in advance the endpoints that will be assessed in the trial, as failure to prespecify endpoints can introduce bias into a trial and creates opportunities for manipulation” [14] (p. 0001). The trial relied on self-reported primary outcomes of fatigue, physical functioning and school presence as measured by questionnaires [11]. The actometer, an objective measurement of activity, was used as a secondary outcome.

Treatment in the study finished after 6 months and the first follow up was at the end of it. A systematic review by Whiting et al. concluded that due to the relapsing nature of CFS, follow-up should continue for at least a further 6–12 months after treatment has ended, to confirm that any improvement observed was due to the intervention and not to the naturally occurring fluctuations of CFS [15]. Furthermore, individuals often experience a decline in therapeutic benefit within weeks after completing CBT [16]. According to Prasad, Cifu and Ioannidis, the use of short-term outcome evaluation affects the credibility of clinical trial results in a negative way [17]. Therefore, LTFU recovery rates of adolescents with CFS, the primary goal of the trial [9], provide a better indicator of treatment success in CFS. 

According to the FITNET authors, web-based CBT consisted of 21 interactive internet-sessions over 6 months [12]. Yet it is not specified (neither in the protocol, nor in either of the two publications), what the exact nature of the intervention in the usual care group was other than that usual care consisted of “individual or group based rehabilitation programmes, cognitive behavioural therapy face to face, or graded exercise treatment, or both, by a physical therapist” [12] (p. 3). Or how many sessions the usual care group participants would/have receive(d). The reason for this given by the authors is that they “could not provide detailed data about the specific interventions in the usual care group because the quality and quantity of cognitive behavioural therapy differed according to local availability” [12] (p. 5). Consequently, 10% of the participants in the usual care group received no treatment at all [12]. Yet as concluded by Coyne, “In evaluating a psychological treatment, it’s important that the comparison/control group offers the same frequency and intensity of contact, positive expectations, attention and support” as the intervention group [18] (p. 1). Otherwise any difference in efficacy of the treatments between the two groups might be simply down to a poorly designed control group. 

Trials of behavioural interventions are unblinded by definition and a number of recent articles have highlighted the problems of erroneous inferences of improvement in their absence, if unblinded trials rely on subjective outcomes [16,19,20]. As noted by Edwards, in rheumatology the use of patient-filled questionnaires was stopped, when they noticed that patients were simply filling in what they thought the researchers wanted to hear [19]. Stouten, who analysed a number of CBT and GET trials that used subjective and objective outcomes, found that the beneficial effect of CBT over specialist medical care vanished when objective outcomes were used [20]. To avoid this, unblinded trials should use objective primary outcomes (alone or together with subjective ones) [16,19,20]. This would have been easily achievable in the FITNET trial had they used the objective actometer not as a secondary outcome but as a primary outcome, alone or together with the (subjective) physical functioning questionnaire.

There are those who take the view that belief in one’s improvement is as good/same as actual improvement. Yet as Stouten concluded, even though “patients *think* they are able to walk more after CBT, they *fail to actually do so*” [20] (p. 3). Subjective improvement without objective improvement means that the treatment is not curative but at best only palliative. 

### 3.2. Patient Selection

The FITNET trial used the Fukuda or CDC criteria [12] which although they are the most widely used CFS criteria, can misdiagnose depressed patients as having CFS. This is because the main characteristic of CFS, namely abnormally delayed muscle recovery after trivial exertion [21], is only an optional criterion and not an essential diagnostic requirement [22] in these selection criteria.

In the FITNET trial, CFS was diagnosed or confirmed by a paediatrician specialising in CFS [12]. During the trial, three patients were given new diagnoses (school phobia, personality disorder or gender identity disorder, and post traumatic stress disorder due to family violence) yet were not excluded from the analysis because of the intention-to-treat-principle [12]. It is important that patients who don’t have the disease being investigated, be excluded from a trial with immediate effect to avoid the erroneous inference of efficacy in its absence, even more so if they have a psychiatric diagnosis in a trial of a behavioural treatment.

In the FITNET intervention group, only 28% of patients had an infection as the triggering factor of their illness [12] even though CFS is also known as post viral fatigue syndrome and classified as such by the World Health Organisation [23]. Tirelli et al., who conducted a study involving 741 CFS patients, found on assessment in their CFS clinic that CFS resulted from prior infectious disease in all patients [3]. Jason et al. studied predictors of CFS in adolescents and found that in 73–78% a mononucleosis-like illness preceded their CFS with almost 50% citing active infection at onset [24].

Also, 21% of patients had a comorbid depression and 14% a comorbid anxiety disorder [12], meaning that up to 35% of patients had comorbid mental health issues, which is almost the same as the 36% of patients classed as ‘recovered’ by the FITNET trial when only one standard deviation was used for their definition of recovery [12]. This is also the very subset of patients one would expect to respond positively to CBT as a meta-analysis by Tolin et al. found cognitive behavioural therapy to be the most effective therapy for anxiety and depression [25].

Reeves et al., co-authored by one of the FITNET investigators, concluded in 2003 that patients with a comorbid medical or psychiatric condition that may explain the chronic fatigue state should be excluded from CFS research studies because overlapping pathophysiology may confound findings specific to CFS [26]. Why the FITNET trial did not do this is unclear. 

### 3.3. Using Primary Outcomes Assessed via Questionnaires 

One of the problems with outcomes assessed via questionnaires in an unblinded trial is ‘response-shift bias’. This occurs when an intervention leads individuals to change their evaluation standard with regard to the dimension measured leading the therapist (and often also the patient) to conclude erroneously that the treatment has worked [16]. This is even more of a problem when the therapy used, in this case internet-based CBT, aims to modify participants’ beliefs and perception of their symptoms.

Other important causes of bias are ‘effort justification’ where patients investing substantial time, energy and effort in an intervention often feel a psychological need to justify this commitment. There is also a tendency for patients/clients to report improvement in accord with what they believe to be the therapist’s/researcher’s hypotheses [16].

Wood et al. in a large scale meta-analysis of over 1300 varied clinical trials, found that in unblinded trials, subjectively assessed outcomes increase the degree of bias and introducing objective outcomes reduces this [27]. Other systemic reviews of clinical trials also concluded that lack of patient blinding combined with self-reporting of outcomes leads to pronounced bias as they become prone to outside influences leading to the erroneous inference of efficacy in its absence, thus making them unreliable [16,28]. Low correlation between objective and subjective activity measurements [29] is not confined to the chronically ill but is also present in the healthy population [30].

### 3.4. The Actometer Results

The Dutch FITNET protocol states that it will be measuring physical performance objectively by using the actometer [11]. However, the results were not reported and the reason for this was not given. Heneghan et al. refer to this as a typical example of “Outcome reporting bias” which “occurs when a study has been published, but some of the outcomes measured and analysed have not been reported”; this “significantly affects the validity” of a study [31] (p. 4).

Wiborg et al. reanalyzed three trials which had reported subjective improvements without reporting their actometer results, and found that CBT didn’t lead to objective improvements [32]. One of the authors of the Dutch FITNET trial [9,12], who is also involved in the NHS FITNET trial [10], was involved in these 3 studies [32].

## 4. Issues with the Conclusions

### 4.1. Long-Term Follow Up (LTFU)

At LTFU, the authors themselves concluded that “usual care led to similar recovery rates” as Internet-based CBT and “~60% of the adolescents recovered, irrespective of the type of treatment” [9] (pp. e1788, e1793) and that “there was no difference between the recovery rates for the different treatment strategies”, yet they went on to say that “The short-term effectiveness of Internet-based CBT on adolescent CFS is maintained at LTFU” [9] (p. e1788). In short the authors ignored the null effect of FITNET at long-term follow-up even though assessing long term recovery was their main goal [9].

### 4.2. Crossing Over 

The Principal Investigator of the upcoming NHS FITNET trial offered this explanation on BBC Radio for the null effect at LTFU: “what happened is that a lot of the children who were in the original control arm” and did not recover then crossed over and “got FITNET as well, so it’s not surprising that at 2 or 3 years, the results were similar” [33] (p. 1). Yet according to the Dutch FITNET trial itself, participants from the usual care group (*n* = 25 at LTFU), who had crossed over to FITNET after the trial ended, showed a recovery rate of 52% at LTFU, compared to 53.6% for those who had not crossed over (*n* = 28 at LTFU). And for those (30.5%) who had crossed over from the FITNET group to usual care, the figure was 61.1%. Thus the authors concluded that “At LTFU, receiving FITNET therapy did not significantly influence recovery rates” [9] (p. 1792), so it made no difference to the recovery rates at LTFU, whether patients had received usual care, FITNET or crossed over from one group to the other. Moreover, as researchers have no control over the environment subsequent to completion of a trial; even if there was a difference, it cannot be attributed to any form of additional post trial treatment. This also takes no account of the carryover effect, whereby the effect of treatment is carried over to the post-treatment completion phase. This is a greater problem in naturally fluctuating diseases like CFS [15]. 

### 4.3. Maternal Concern

The authors stated that a factor related to recovery at LTFU was “maternal focus on bodily symptoms” and concluded that it “suggests that an intergenerational vulnerability and interaction between mother and child may exist” [9] (p. e1794). Yet it is normal for a parent to worry about (the symptoms of) her ill child when there’s no recovery or improvement and maternal concern is a variable that will itself be influenced by how ill the child is and concluding that it increases illness duration is confusing cause with effect.

## 5. Issues with the Definition of Recovery

### 5.1. Naturally Occurring Recovery 

FITNET didn’t take the naturally occurring recovery process in adolescents with CFS, which has been widely reported in scientific literature, into account. According to Joyce et al. and Patel et al., 54–94% and 80% of adolescents respectively, recover spontaneously within 3–4 years of disease onset [34,35]. Bell concluded that up to 47% of children and adolescents with CFS spontaneously recover and a further 27–46% improve [36], and Krilov et al. found that at LTFU, 43% of families considered their child “cured” and 52% considered their child “improved,” whereas only 5% considered their child to be “the same” [37]. Bell et al. (a study cited by the FITNET trial) found that at LTFU 80% of adolescents/children reported spontaneous remission from the disease [38]. And in a prospective study, Feder et al. found that at follow-up with a mean of 3.8 years, 65% of adolescents had recovered and a further 29% had improved, both spontaneously [39]. This is similar to the findings by Rangel et al. who found that at a mean follow-up of 38 months, 67% of severely affected patients had recovered [40] as can be seen in Table 1. And finally, a recent study by Norris et al. (co-authored by the principle investigator of the NHS FITNET trial) entitled “Natural course of chronic fatigue syndrome/myalgic encephalomyelitis in adolescents”, concluded that approximately 75% of adolescents who do not receive treatment will recover after 2–3 years [41].

As Table 1 shows, spontaneous recovery percentages at LTFU are similar to or better than the FITNET LTFU results (mean average of 4.5 years post illness onset) of 63% and 60% in the groups receiving the FITNET treatment and usual care, respectively [9]. This shows that contrary to the published conclusions of the trial, neither internet-based CBT, nor usual care lead to an increase in the percentage of adolescents who recover spontaneously, suggesting that neither are effective. 

### 5.2. The Definition of Recovery

According to the FITNET trial, “Recovery was defined post hoc in relation to healthy peers (±2 SD)” [12] (p. 3), and was defined as a “combined end point” consisting of a “fatigue-severity score <40, a physical functioning score of ≥85%, school/work presence of ≥90%, and an SRI answered with (1) “I have completely recovered” or (2) “I feel much better.”” [9] (p. e1791).

Post hoc means that the recovery criteria were defined after the trial’s end, contrary to normal practice where endpoints are defined prior to trial commencement [14]. This means that the results will influence the recovery definition and this as pointed out by Goldacre “allows the … ‘random error’ in your data to exaggerate your results (or even yield an outright false positive, showing a treatment to be superior when in reality it’s not)” leading to the wrong conclusions and “in medicine, that’s not a matter of academic sophistry—it causes avoidable suffering” [45] (p. 1). And according to Ioannidis: “Flexibility increases the potential for transforming what would be “negative” results into “positive” results”, and “The greater the flexibility in designs, definitions, outcomes, and analytical modes … the less likely the research findings are to be true” [46] (p. 0698). 

The flexibility in post-hoc defining of recovery is high and using a post hoc recovery definition enabled the authors to tailor make this to fit the results. They then chose to use a very wide recovery definition instead of a narrow/tight one reflecting actual recovery. For example, the trial inclusion criteria had a “Fatigue severity subscale (CIS-20) score ≥40” and a “Physical functioning (Child Health Questionnaire) score <85” (out of 100) [11] (p. 3). Yet the recovery criteria were “a fatigue-severity score <40” and “a physical functioning score of ≥85%” [9] (p. e1791) so that with minimal improvement from 40 to 39 on the fatigue scale or from 84 to 85 on the physical functioning scale, one could be classed as recovered according to that particular outcome. A score of 83 (out of 100) denotes a “good outcome, as it represents the ability to carry out moderate activities (e.g., carrying purchases, moving furniture)” [47] (p. 2039), so that patients with scores of 80 to 85, were already well on the way to recovery and only mildly affected, yet they were still ill enough to enter the FITNET trial. A patient feeling much better has improved, not recovered yet and should not be classed as such; by using that as part of their recovery definition, the authors labelled patients still ill, as recovered.

In the FITNET study there was also an overlap between recovery and severely fatigued as illustrated by the following: “Fatigue was measured with the subscale fatigue severity of the CIS-20 (range 8–56)” [12] (p. 3) and a patient with a score of less than 40 was considered recovered. Yet according to the literature, severe fatigue is defined as a CIS-fatigue severity score of greater than or equal to 35 [48,49,50,51,52]. Therefore, a fatigue score between 35 (inclusive) and 40, meant recovered and severely affected at the same time. Furthermore, according to Kennedy, recovery “is the elimination of … symptoms and a return to premorbid levels of functioning” [53] (p. 233) which the FITNET trial ignored. That the 63% of patients reported as recovered might have included those who had improved rather than recovered is indirectly acknowledged by the authors in their appendix where they stated that had they created a definition of recovery using 1 rather than 2 standard deviations, the recovery rate would have been much lower (36% rather than 63%) [12]. 

As noted above, a score of 83 (out of 100) “represents the ability to carry out moderate activities (e.g., carrying purchases, moving furniture)” [47] (p. 2039) and a score of 100 represents the ability to do all activities according to van Geelen et al., a study co-authored by one of the FITNET investigators [54]. Yet in FITNET, a physical functioning score of ≥85% [9] was sufficient to be classed as recovered. Also, according to Bowling et al., healthy 18.5 year olds (the mean age of trial participant at LTFU in the Dutch study) [9], have a physical functioning score of 100 [55]. And as their scores are not evenly distributed but highly skewed, with nearly everyone in the maximum range of 100% [55], this will cause standard deviations to be grossly inflated, making them an unsuitable measure of variability and inappropriate for use [56].

In a study which found that exercise causes immunological abnormalities in CFS, White et al. found that sedentary controls with a mean age of 38 had a physical functioning score of 100 (out of 100) and a Chalder Fatigue score of 0 [57]. So one would expect the physical functioning of healthy 18.5 year olds to be at least the same, if not better than that of healthy sedentary 38 year olds, who have the minimum scores for healthy people of that age. Yet the Dutch study described a physical functioning score of 85% or more as ‘recovered’.

### 5.3. School/Work Attendance

With regards to school attendance, presence of ≥90% [9] was required to be classed as recovered. School presence at the end of treatment, i.e., at 6 months, was measured by looking at the two previous weeks, which is too short a time window to make a meaningful conclusion about recovery. In their 2012 paper, the authors state that “A particular strength is that the main outcome (school attendance) was checked and double checked by the investigators, parents, teachers, and therapists” [12] (p. 6), yet in the same article they noted that “On the day of testing, the past 2 weeks of school attendance were validated with a general questionnaire and checked with the parents” [12] (p. 3). In their long-term follow-up paper, they state that “At LTFU, school presence was scored by using retrospective questionnaires. These are less precise than the prospective diaries used during the FITNET trial and at the 12-month assessment” [9] (p. e1793). In other words, school presence was scored by using retrospective questionnaires at all 3 different checkpoints during the trial; there was hardly any check on attendance at the 6 and 12 month checkpoints (i.e., at the end of treatment and six months later), and at LTFU school attendance was reported retrospectively by the patients without any verification by the school at all [9]. As up to 35% of patients in the trial suffered from anxiety and/or depression [12] this predisposed them to selective recall, which added to the fact that participants often want to please researchers, might further influence the reliability of retrospective reporting of school attendance via questionnaires [16].

In addition, patients with CFS often have concentration and memory problems [58], so school attendance does not translate to school performance as highlighted by Patel et al., who found that one third of adolescents who were back at school full time did not get a qualification [35] as they had improved enough to attend school full time but not enough to study at their premorbid levels. Nijhof et al. found that adolescents with CFS suffer from neurocognitive impairment and “that the actual lower IQ levels in CFS adolescents represent a consequence rather than a cause of CFS” and went on to conclude that neurocognitive impairment and “IQ scores should normalize after treatment of and recovery from CFS” [59] (p. 250). However, Knoop et al. found that CBT does not lead to objective improvement on neuropsychological testing [58].

At long term follow up, 30% of FITNET participants had left school [9]. Yet only 30% of them were working full-time [9]. This indicates that even though FITNET classed 63% as recovered, patients hadn’t recovered enough to be working full-time, i.e., had not actually recovered. 

Work absence of 10% (or less) is 26 days of sick leave per year. The average number of days of sick leave in the Netherlands is 7 (in Ireland and the UK it is even lower at 4) [60]. This is a school or work attendance rate of nearly 100% (97% or higher), for the average working-age population (those aged 18–67, including the chronically ill). Yet this trial regarded an attendance rate of 90% or more at LTFU as recovery. Moreover, 6% of participants in the treatment group of the trial already had a school/work attendance of 85% or more at trial commencement [12]. Therefore, these participants would have already been considered to be nearly or fully recovered—according to 1 of the 4 recovery criteria upon entering the trial—before receiving any treatment at all. 

## 6. Issues with the NHS FITNET Trial

### 6.1. Background Information 

The UK NHS FITNET trial which will cost nearly £1 million [10], is based on the ‘success’ of the Dutch FITNET trial, with three of the authors/investigators of the latter [9,12] involved in its design and execution [10,61]. 

The main investigator of the NHS FITNET trial claimed, in a widespread media campaign, that internet CBT is a successful treatment for 2/3 of adolescents with CFS [62] before the trial had commenced or participants had been recruited.

According to the protocol, the UK NHS FITNET trial is a large randomised controlled trial involving 734 patients, divided into two groups “to investigate whether CBT specifically designed for CFS/ME and delivered over the internet … is effective and cost-effective compared to Activity Management for children with CFS/ME who do not have access to a local specialist CFS/ ME service” [61] (p. 7). To be included in the trial children need to be aged 11 to 17 years, have “CFS/ME (defined using NICE guidance … with no local specialist CFS/ME service” [61] (p. 10).

According to the protocol “There is good evidence that CBT is effective” and “In particular, the PACE trial showed that both CBT and GET were...effective” [61] (pp. 6, 7), whereas independent reviews of the PACE trial showed that these treatments are not effective [63,64].

The protocol further states that “Those children with 3 months of disabling fatigue plus one symptom (Nice guidance) will be eligible. The NICE guidance will be used and not the Centre Disease Control (CDC) criteria, based on the relevance of NICE criteria to the NHS” [61] (p. 11). Yet according to the NICE guidelines, the fatigue needs to include all of the following features: “new or had a specific onset (that is, it is not lifelong); persistent and/or recurrent; unexplained by other conditions; has resulted in a substantial reduction in activity level; characterised by post-exertional malaise and/or fatigue (typically delayed, for example by at least 24 h, with slow recovery over several days)” [65] (pp. 14–15). The NICE guidelines also require patients to have at least one of ten listed symptoms. 

### 6.2. Protocol Design, Diagnosis and the Control Group

The NHS FITNET website, under the Frequently Asked Questions, states that “One of the most helpful symptoms to make a diagnosis is “post exertional malaise” which is an increase in fatigue (and other symptoms) after exertion” [66] (p. FAQs), leading the reader to assume that they are following the NICE guidelines, when in fact they are not. Not having PEM as a prerequisite, and selecting “children with” ‘just’ “3 months of disabling fatigue” [61] (p. 11), means that the NHS FITNET trial has no way of ensuring that the patients have indeed CFS and not just chronic fatigue. 

The researchers themselves state that: “The second ethical issue is that we need to be certain that this trial recruits children with CFS/ME and not with other disorders that present with fatigue”. “We have therefore put in place rigorous assessments to ensure that other causes of fatigue are diagnosed and referred for appropriate treatment” [61] (p. 25). “There is a small risk that the trial may recruit children that do not have CFS/ME but instead have other disorders that present with fatigue” [61] (p. 26). However, by omitting PEM (the hallmark symptom of the disease), that risk is not small but in fact large. It’s akin to doing research into the treatment of fractures but omitting the requirement of having a broken bone.

Please note that some changes were made to version 3.0 of the NHS FITNET protocol [61] after we had submitted our paper for publication. One of these changes, namely the inclusion of post exertional malaise (PEM) as a compulsory diagnostic requirement in the latest version of the protocol [67], alleviates some of our concerns about patient selection. However it is unclear why this protocol change was made at this stage or why PEM wasn’t included in the original protocol in accordance with the NICE guidelines being used.

According to the NHS FITNET protocol, participants assigned to internet-based CBT will receive 19 sessions lasting 60 min over a six-month period while those in the activity management group “will have up to three video (e.g., Skype) appointments (one assessment and two follow up)” from an occupational therapist from the trial (after which the participant’s nominated local therapist or doctor is asked for a “review within six to eight weeks”) [61] (p. 13). Yet in evaluating a psychological treatment, it’s important that the control group offers the same frequency and intensity of contact, positive expectations, attention and support as the intervention group otherwise any difference between the two groups might be simply down to the design of the study leading to the erroneous inference of efficacy in its absence [18]. 

The protocol doesn’t state the maximum entry score for their primary outcome nor does it define improvement or recovery.

Consequently, this trial, like other CBT trials is an unblinded trial by definition which relies on one subjective outcome. Yet as mentioned above, unblinded trials should use objective primary outcomes (instead or as well), to safeguard against erroneous inference of efficacy in its absence [16].

### 6.3. Illness Fluctuations

The authors themselves state that: “CFS/ME is by its nature, a fluctuating illness” [61] (p. 22), yet their primary outcome (disability measured using the Physical Function Scale (SF-36-PFS)) is directly at the end of treatment (at 6 months) and not as a systematic review by Whiting et al. advised, at least 6–12 months after a trial ended, to eliminate the effects of these fluctuations [15]. 

### 6.4. Activity Management

The protocol states that “Activity Management is used as the comparator in this study as it is recommended by the National Institute of Health & Care Excellence (NICE)” [61] (p. 5). The trial’s protocol describes activity management in the following manner: “When participants have managed the baseline for 1–2 weeks, they will be asked to increase this by 10–20% each week” “until they are able to do up to 8 hours of activity a day” [61] (pp. 13–14). Yet according to the NICE guidelines ‘activity management’ is “A way for people to manage their symptoms by learning to analyse and plan activities so that they can achieve more at home, at work and at leisure” [68] (p. 1) with “Gradually increasing activity above the baseline in agreement with the person” [65] (p. 31). This is how the NICE guidelines describes GET: “When the low-intensity exercise can be sustained for 5 days out of 7 (usually accompanied by a reduction in perceived exertion), the duration should be reviewed and increased, if appropriate, by up to 20%” [65] (p. 29). So it seems that the NHS FITNET trial isn’t using activity management but a form of GET by instructing participants to increase activity by “10–20% each week” [61] (p. 14), irrespective of symptoms. Patient surveys by ME Associations over the last two decades have consistently shown that patients pushing beyond their limits will trigger relapses [5,6,7,8]. Meeus et al., who used an actometer to assess activity, found a close link between symptom exacerbation and physical activity and concluded that one needed to be extremely careful using physical activity in patients with CFS [69]. Many patients who are bedridden with severe ME were not severely affected before being treated with CBT and GET [70]. Objective evidence provided by Paul et al. and Black and McCully confirmed that CFS patients suffer from delayed recovery and worsening of symptoms following exercise [71,72]. A recent study by Cook et al. provided objective evidence of “the detrimental neurophysiological effects of post-exertion malaise” caused by exercise which “exacerbated symptoms, impaired cognitive performance and affected brain function in Myalgic Encephalomyelitis/Chronic Fatigue Syndrome patients” [73] (p. 87). Moss-Morris et al. reported that in 40% of patients their health deteriorated as a consequence of exercise [74]. A trial by Núñez et al. found that treatment with CBT and GET led to “worse SF-36 physical function and bodily pain scores” [75] (p. 381). A review of the use of CBT and GET in the Belgium government CFS Centres showed that treatment with CBT and GET did not improve physical capacity, lowered employment status and the “percentage of patients living from a sickness allowance increased” [76] (p. 80). 

### 6.5. The Boom and Bust Theory

According to the NHS FITNET protocol, “The description of activity and function in CFS/ME is one of boom-bust which usually occurs over several days and sometimes weeks. “Payback” or “crashes” or “flares” are to be expected in young people whether or not they are undergoing treatment. Payback, crashes or flares can mean that a child who was previously mobile becomes bed-bound or is unable to go to school. Episodes can last days or occasionally weeks. Treatment is designed to reduce these over time.” And “Any adverse event will be defined as a serious adverse event if: it results in death, is life threatening, requires hospitalisation or prolongation of existing hospitalisation, results in persistent or significant disability or incapacity” [61] (p. 22).

The NHS FITNET trial authors are using their ‘boom and bust’ theory of CFS to blame any deterioration on the disease, and not on the adverse effects of their own treatment. This was disproved by Van der Werf et al. (co-authored by an investigator also involved in both the Dutch and the UK NHS FITNET trial) [77] and in a systemic review by Evering et al. [78]. Using or re-analysing actometer readings, both found no significant differences during the day or in the day-to-day fluctuations in activity patterns between controls and CFS patients. 

## 7. Discussion

The FITNET trial, which like all CBT trials, was unblinded by definition, involved 135 adolescents with CFS aged 12–18 years at the start of the trial. It compared the effectiveness of internet-based CBT with that of usual care and used questionnaires to assess its primary outcomes (fatigue severity, physical functioning and school attendance). The trial concluded that after six months, i.e., at the end of treatment, 60% had recovered in the Internet based CBT group and only 8% in the usual care group. It also concluded that at LTFU (mean of 2.7 years after treatment’s start) the positive effects of FITNET were maintained, 58.9% of adolescents had recovered from CFS and usual care led to similar recovery rates, although these rates were achieved at a slower pace [9].

There were a number of important issues with the trial including selection of the patients and the design of the control group. For instance, 72% of participants in the intervention group did not have a preceding viral illness/infection casting doubt on the validity of their CFS diagnosis.

Also patients in the intervention group received 21 sessions of internet-based CBT, yet the exact nature of the treatment nor its intensity in the usual care group were specified to the point that 10% of participants in this group did not receive any treatment at all, ignoring the fundamental tenet of proper trial design that participants in the control group receive the same level of treatment, care and attention as those in the treatment group.

The differences at the end of treatment might also be due to response shift bias or patients filling in questionnaires in such a way as to please the investigators. This is why rheumatology trials stopped the use of questionnaires in favour of using objective outcomes [19]. The FITNET trial did not publish the actometer results. This is a form of outcome reporting bias that endangers the validity of a trial [31]. Reanalysis of three other trials all involving one of the FITNET trial investigators, that did not publish their actometer results either, showed that CBT did not lead to objective improvement [32].

In summary, regarding outcome differences at 6 months, it’s highly likely that the combination of an unblinded trial, subjective outcomes and a poorly chosen control group, alone or together with response shift bias and/or patients filling in questionnaires in a manner to please the investigators produced the appearance of positive effects, despite the lack of any substantial benefit to the patients, leading to the erroneous inference of efficacy in its absence.

Furthermore, there was no difference in recovery rates between the two groups at LTFU, the trial’s primary goal. Crossing over from internet-based CBT to usual care or vice versa, did not influence recovery rates. The trial did not take into account the widely reported spontaneous recovery in adolescents with CFS (Table 1) [35,36,37,38,39,40,41,42,43,44], which is similar to or better than the reported recovery rates in this trial. That is without taking into account the problems with the post hoc definition of recovery used by the FITNET trial which was tailor-made to fit the results yet was so wide that it included the severely fatigued/ill. The trial acknowledged that had they used a definition of recovery with one standard deviation instead of two, their recovery rate would drop from 63% to 36%. But even this definition didn’t resemble actual recovery as defined by Kennedy, where symptoms are gone and patient health returns to pre-illness levels [53]. This suggests that Internet based CBT and also usual care consisting of individual or group based rehabilitation programmes, CBT or graded exercise treatment, or both, might actually impede the naturally occurring recovery process in adolescents with CFS. Moreover, up to 35% of FITNET patients (similar to the 36% if only one standard deviation had been used), had a comorbid depression and/or anxiety disorder for which CBT is the most effective treatment according to a meta-analysis by Tolin et al. [25]; confounding the results of a trial that used a behavioural treatment. Reeves et al., co-authored by one of the FITNET investigators, concluded in 2003 that these patients should be excluded from CFS research studies [26].

The UK NHS has allocated almost a million pounds to replicate the Dutch FITNET trial based on the reported recovery rates at the end of treatment. It is likely that had they been aware of the above issues of the Dutch FITNET trial, they might have reconsidered this. As the 2 trials are almost identical, the current NHS trial will also suffer from the above-mentioned methodological problems of the Dutch study.

In conclusion, it is paramount that the general public as well as patients and health professionals, be aware of the proclamations of outcome success and high recovery rates from psychological interventions in the absence of objective data supporting these claims, especially when patients repeatedly testify that these treatments are ineffective and even harmful [5,6,7,8]. Our reanalysis of the FITNET trial showed that this trial suffered from severe methodological problems including using a definition of recovery that included the severely ill.

## 8. Strengths and Weaknesses of the FITNET Trial

A particular strength of the trial was that they did not rely exclusively on subjective outcomes but also used the actometer, an objective measure of activity. The second strength of the study was their primary goal of recovery at long-term follow-up, which removes the effects of the naturally occurring fluctuations of CFS, according to the systemic review by Whiting et al. [15].

Its weaknesses were:
*a protocol that was published years after the trial started plus a badly designed control group;*patient selection issues such as more than 70% of those in the treatment group not having an infectious onset;*not excluding participants with a comorbid psychiatric disorder or those who during the trial were found not to have CFS but a psychiatric disorder instead;*a post hoc definition of recovery that included the severely fatigued/ill, yet labelling it as a strength of the study;*not publishing the actometer results, thus causing reporting bias;*not taking into account the naturally occurring recovery rates in adolescents with CFS as documented by many studies;*ignoring its own primary goal (recovery rates at LTFU) as well as the null effect at LTFU.

## Figures and Tables

**Table 1 behavsci-07-00052-t001:** Proportions of spontaneously recovered and improved adolescents with Chronic Fatigue Syndrome.

Reference	Recovered	Improved	Mean LTFU
Marshall et al. 1991 [42]	-	76% definite improvement	2.2 years
Feder et al. 1994 [39]	65%	29%	3.8 years
Bell 1995 [36]	up to 47%	27–46%	26.7 months
Joyce et al. 1997 [34]	54% to 94%	-	18 to 38 months
Krilov et al. 1998 [37]	43% (“cured”)	52%	1–3 years
Rangel et al. 2000 [40]	67%	-	3.8 years
Bell et al. 2001 [38]	37.1%	42.9% greatly improved (well but not resolved)	13 years
Patel et al. 2003 [35]	80.6% recovered/improved	-	2.5–3 years
Gill et al. 2004 [43]	25% showed near to complete improvement	31% showed partial improvement	4.57 years
Norris et al. 2017 [41]	75%	-	2–3 years
Rødevand 2017 [44]	most recover	-	3 years

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
