# Peer review of "FITNET’s Internet-Based Cognitive Behavioural Therapy Is Ineffective and May Impede Natural Recovery in Adolescents with Myalgic Encephalomyelitis/Chronic Fatigue Syndrome. A Review"

_behavsci, 2017, doi:10.3390/bs7030052_

Round 1

Reviewer 1 Report

Abstract:

The authors write “This  has  serious  implications for the NHS FITNET trial which is a blueprint of the Dutch study as it  doesn't  make  sense  economically  or  scientifically  to  duplicate  such  a  trial.  Furthermore, it’s  unethical  to  expose  children/adolescents  to  ineffective  and  potentially harmful treatments.”

-          Please rewrite or remove this sentence. Replication is a corner stone of science – you can argue problems with FITNET Holland may have implications and if the same study design is used in the UK the same biases might creep in – but you can’t argue studies can’t be replicated. Also, ethics is a broad field, you can’t make sweeping statements, you must state what the ethical objections are. In both FITNET and the UK study the authors would argue that their studies passed ethics approval – you may have an issue with this, but their defence is solid – so you must specify your objection supported with facts; hence I can’t allow such a sweeping statement.

Page Length – 22 pages is rather long for a paper that simply reanalyses one study to demonstrate potential flaws and biases. I would urge you to consider trying to shorten the manuscript because while I may be familiar with the literature and have an ability to wade through this paper, may readers will be either put off by the length or will lose focus as one moves from page to page – best to keep a short word limit and stick to key points in as concise a way as is feasibly possible. No reader or reviewer wants to spend hours reading a paper – this is the point really, make it a digestible size.

A  recent  independent  review  of  the  PACE  trial  54 showed that cognitive behaviour therapy (CBT) and graded exercise therapy (GET)  55 are  ineffective  [9]  and  further  separate  reanalyses  of  the  raw  data  showed  that  56 neither lead to actual recovery [10,11].

-          You don’t demonstrate balance here – PACE authors published 16 papers suggesting some benefits, including some recovery for some patients, even in the control groups – I understand reanalysis showed falling rates, you need to balance with what the PACE authors said they found v what reanalysis found.

  Helmfrid recently concluded that using  58 CBT and GET for ME/CFS is not evidence based and "graded exercise therapy often  59 leads to deterioration" [13] (p. 7)

Grammar?

Both  68 authors of this article have severe long term ME and have found both CBT/GET not  69 only ineffective but in fact harmful.

..this is irrelevant and introduces strong personal bias – remove

Line 74 – I think you are misinterpreting what the BMJ called for – patients to be involved at all stages. I would remove those lines next 3-4.

L94 – when trial was submitted and accepted – irrelevant details

L100 – page 0001?

L104 – you claim FITNET is not an RCT – I agree it fails to meet many of the tenets of an good RCT but the authors might argue that many psycho-behavioural interventions do not blind participants and use alternative treatments as controls, rather than traditional placebo control. All you can argue is it’s a poor RCT, or that the Control is missing – there is randomisation, there is treatment – so the C missing in the R-C-T.

L112 – you talk about objective measures without defining what is an objective measure v what is a subjective measure – and here you will struggle, for instance defining objectivity, also subjective measures are relevant and important too, you say no objective measures – are you sure, can you list what measures were used and give a good argument why they should not be classed as objective?

L113 – self reported school attendance is meaningless – it might be weaker than other forms of measurement but it’s not “meaningless”.

I am afraid at this stage of review I need to send paper back to authors to ask them to consider the language the use throughout the paper. It is important the authors take a neutral stance and present the evidence in a balanced and fair manner and use terms correctly and proportionately – currently it would take hours and hours to wade through the manuscript to correct what appear to be problems every few lines that I would need corrected in order to pass this paper.

It is unfortunate the paper is not better written and presented as I wholly concur with the authors that the evidence presented in the FITNET trial is questionable, the methods used and the interpretation of results/outcomes requires analysis and attention and I commend the authors for tackling this task – however they must present a professional and strong case for a paper to be published – I as reviewer would be uncomfortable passing this paper in its current format and I don’t have the time or inclination to spend a day guiding the authors through a re-write – that’s their job.

I would ask them to spend a few days or a week or two going through the paper – also I would strongly urge them to get support from a senior academic who might guide them regarding the types of sweeping statements they make in this paper. Their paper has merit and the authors should not be discouraged by my comments around style – however it’s incumbent on me as a reviewer to be fair in review, equitable and impartial, which means despite the fact I agree with a central trust of this paper and its findings, it’s just too poorly written for me to accept at this stage.

Abstract:

The authors write “This  has  serious  implications for the NHS FITNET trial which is a blueprint of the Dutch study as it  doesn't  make  sense  economically  or  scientifically  to  duplicate  such  a  trial.  Furthermore, it’s  unethical  to  expose  children/adolescents  to  ineffective  and  potentially harmful treatments.”

-          Please rewrite or remove this sentence. Replication is a corner stone of science – you can argue problems with FITNET Holland may have implications and if the same study design is used in the UK the same biases might creep in – but you can’t argue studies can’t be replicated. Also, ethics is a broad field, you can’t make sweeping statements, you must state what the ethical objections are. In both FITNET and the UK study the authors would argue that their studies passed ethics approval – you may have an issue with this, but their defence is solid – so you must specify your objection supported with facts; hence I can’t allow such a sweeping statement.

Page Length – 22 pages is rather long for a paper that simply reanalyses one study to demonstrate potential flaws and biases. I would urge you to consider trying to shorten the manuscript because while I may be familiar with the literature and have an ability to wade through this paper, may readers will be either put off by the length or will lose focus as one moves from page to page – best to keep a short word limit and stick to key points in as concise a way as is feasibly possible. No reader or reviewer wants to spend hours reading a paper – this is the point really, make it a digestible size.

A  recent  independent  review  of  the  PACE  trial  54 showed that cognitive behaviour therapy (CBT) and graded exercise therapy (GET)  55 are  ineffective  [9]  and  further  separate  reanalyses  of  the  raw  data  showed  that  56 neither lead to actual recovery [10,11].

-          You don’t demonstrate balance here – PACE authors published 16 papers suggesting some benefits, including some recovery for some patients, even in the control groups – I understand reanalysis showed falling rates, you need to balance with what the PACE authors said they found v what reanalysis found.

  Helmfrid recently concluded that using  58 CBT and GET for ME/CFS is not evidence based and "graded exercise therapy often  59 leads to deterioration" [13] (p. 7)

Grammar?

Both  68 authors of this article have severe long term ME and have found both CBT/GET not  69 only ineffective but in fact harmful.

..this is irrelevant and introduces strong personal bias – remove

Line 74 – I think you are misinterpreting what the BMJ called for – patients to be involved at all stages. I would remove those lines next 3-4.

L94 – when trial was submitted and accepted – irrelevant details

L100 – page 0001?

L104 – you claim FITNET is not an RCT – I agree it fails to meet many of the tenets of an good RCT but the authors might argue that many psycho-behavioural interventions do not blind participants and use alternative treatments as controls, rather than traditional placebo control. All you can argue is it’s a poor RCT, or that the Control is missing – there is randomisation, there is treatment – so the C missing in the R-C-T.

L112 – you talk about objective measures without defining what is an objective measure v what is a subjective measure – and here you will struggle, for instance defining objectivity, also subjective measures are relevant and important too, you say no objective measures – are you sure, can you list what measures were used and give a good argument why they should not be classed as objective?

L113 – self reported school attendance is meaningless – it might be weaker than other forms of measurement but it’s not “meaningless”.

I am afraid at this stage of review I need to send paper back to authors to ask them to consider the language the use throughout the paper. It is important the authors take a neutral stance and present the evidence in a balanced and fair manner and use terms correctly and proportionately – currently it would take hours and hours to wade through the manuscript to correct what appear to be problems every few lines that I would need corrected in order to pass this paper.

It is unfortunate the paper is not better written and presented as I wholly concur with the authors that the evidence presented in the FITNET trial is questionable, the methods used and the interpretation of results/outcomes requires analysis and attention and I commend the authors for tackling this task – however they must present a professional and strong case for a paper to be published – I as reviewer would be uncomfortable passing this paper in its current format and I don’t have the time or inclination to spend a day guiding the authors through a re-write – that’s their job.

I would ask them to spend a few days or a week or two going through the paper – also I would strongly urge them to get support from a senior academic who might guide them regarding the types of sweeping statements they make in this paper. Their paper has merit and the authors should not be discouraged by my comments around style – however it’s incumbent on me as a reviewer to be fair in review, equitable and impartial, which means despite the fact I agree with a central trust of this paper and its findings, it’s just too poorly written for me to accept at this stage.

this paper has important material regarding an important subject - that of evidence derived from a large randomised controlled trial of treatments for children with chronic fatigue syndrome. The authors present some interesting and important findings from their analysis on one trial - FITNET - and I believe this paper is important, but it is poorly written in places and makes far too many sweeping statements and it fails to present a balanced and neurtral viewpoint - which detracts from the important findings within. I would recommend a re-write at this stage.

Author Response

Thank you for reading our article and your comments.

We have shortened the article, and stuck to key points in as concise a way as is feasibly possible; changed the tone so that it is as neutral as possible; made the intro neutral, the article easier to read and reorganised our points based on themes (i.e. issues with patient selection as suggested by reviewer 2);

As suggested we have removed the part from the abstract the reviewer objected too. And the same for the sentences/parts he objected too.

We have added a section about strengths and weaknesses of the trial as requested by reviewer 2.

Reviewer 2 Report

Thank you for giving me the chance to read this manuscript. The key ideas and arguments in this work are worth bringing to the attention of readers. However, I also think the manuscript could be tightened in various ways that would significantly tighten the arguments and increase its impact. Below I set out some recommendations for revision.

Introduction

The introduction needs to be shortened. You want to make this smother and easier to read so the reader is motivated to press on to the end. Keep it to the basics: provide a description of the condition known as CFS, based on the most commonly used case definition(s), and summarise the existing research regarding similar behavioural treatments. You could also mention one or two key research findings regarding CFS (e.g., physiological abnormalities following exertion), but be brief. Avoid theories, speculative statements, and direct quotations, unless they pertain to the study you are about to describe. Don’t pre-empt your conclusions by discussing in detail other people’s opinions on these treatments.

Above all, you want the intro to be neutral - it should be written so that the authors’ opinion of the trial is not yet obvious.

The trial under examination

The ms needs to include a section that summarises the main aspects of the trial under examination, and that does so in a completely neutral, non-evaluative way. This section would briefly describe the authors’ rationale for conducting the trial, the trial, design, the key aspects of the methods, the results and the researchers’ conclusions. All critical analysis of the trial should be left to subsequent sections.

Critical analysis: Tightening your arguments

The critique section advances some arguments that are potentially strong. But the citation of supporting evidence needs to be tightened. You need to make clear which referenced statements refer to actual evidence, and which ones refer to someone else’s viewpoint or conclusion. So for example, the ms cites a conclusion from a study by Wood et al (that unblinded trials with subjective endpoints may produce overoptimistic results) and one from a study by Song and Jason (“the model of preoccupation with bodily symptoms … was not valid”). It would be much stronger to describe the actual evidence on which these conclusions were based. Instead of saying what the authors concluded, describe what their actual findings were.

This is all about showing consistently that you appreciate the distinction between actual findings and the conclusions that researchers draw from them. The latter can always turn out to be wrong – and that’s the very core of your criticism of the FITNET trial.

Another place you could tighten is in the critique of the CFS case definition used in FITNET. The section says “The fact that ME/CFS is known as a post-infectious disorder.” To make a strong case here, you don’t want to rely on what other people believe, argue or think. Cite primary evidence - the proportion of post-infectious cases in other published cohorts, and how measured (whether participants asked to recall, or what?). If the cohort selected in the FITNET trial differed in some important respects from the others, then you might call into question whether the results generalise. A lot of the relevant evidence is already in the ms – it’s just a matter of making the nature of the argument clear to the reader.

Specific points of argument

The points about illness fluctuation and spontaneous recovery are good ones. However, you need to explain why the usual care group did not control adequately for those factors. Is there any reason why we would expect more rapid spontaneous resolution in the patients who were selected into the trial than in those in the usual care group? Can you mount an argument of that nature?

There are good points here about the trial’s odds conclusions on the maternal concern issue. We cannot conclude that maternal concern for their child’s health increases illness duration, because maternal concern is a variable that will itself be influenced by how ill the child is. So this is a classic case of confusing cause with effect.

The points about the definition of recovery are also strong. The ms authors point out clearly why the definition in the trial is inadequate.

I did notice that there is no consideration of strengths of the trial. There must have been some qualities to the trial that are worth a quick mention? It is important to provide balance, even if your final conclusions are that the weaknesses far outweigh the strenghts.

I have one more suggestion, which the authors can accept or reject as they wish. I suggest avoiding the term “placebo effect” altogether, and instead describing the various phenomena that may contribute to spontaneous improvement in a (blinded or unblinded) control condition. As you are aware, these include lots of things aside from expectation effects, including spontaneous recovery, natural fluctuations in the illness course, regression to the mean (since patients often seek treatment when their symptoms are at their worst). You could perhaps bring in the term “illusory placebo effect” when referring to the expectation-induced component – but be clear that this is a reporting artefact only, and does not necessarily confer any enduring benefit. There are people who take the view that belief in one’s improvement is as good as – or the same as – actual improvement.

Tone

The impact of the entire article would be stronger if the tone were more neutral. For example, on line 119 onwards on p. 3: “…telling patients that ignoring their symptoms …will make them better [20].” Try rewording this in a way that does not reveal your own contempt quite so transparently!  When describing the “unproven ‘boom or bust’” theory (line 718, p. 18), simply call it the “boom or bust” theory, and leave out the word unproven.

Another example: at line 602 onward, p 15: “This totally ignores the outcome of the independent reviews of the PACE trial which showed that the conclusion based on the PACE trial's published results should have been that these treatments are ineffective”. Remove the word “totally”, it betrays your feelings! Also, I would suggest probably avoiding the issue of whether the PACE treatments were effective (that’s a discussion in its own right that requires citing proper evidence and arguments). Better to stick to FITNET. Other trials are relevant only in so far as they inform about FITNET. Keep your gunpowder dry for the really critical points you want to make.

Avoid speculating on the trial authors’ intentions. Line 561, p. 14 states, “Their failure to do so suggest that they found no objective improvement, and chose not to report this”. If you state the facts clearly, people can draw their own conclusions as to why these results were omitted. It is okay to say that the authors did not provide sufficient justification for omitting these results (if that is indeed the case), but avoid inferring what you think their intentions were in not providing these data.

Finally, even though your own perspective as patients is valuable (and will have given you some unique insights into the problems of this trial), I would be disinclined to refer to your personal experience explicitly in this peice. Drawing on more objective sources of evidence will be much more persuasive.

Organisation

If you want people to read this, make the revised version shorter – remove points that are not central, so as to highlight those points that you really want the reader to grasp.

Then, when you’ve pared down your analysis to those key points, have another think about the best way to organise your points – you could do it based on themes (e.g., issues with patient selection, issues with outcome measurement issues relating to the non-adherence to protocol etc.), or consider first strengths then weaknesses. Then I would use a more systematic subheading system, so the reader really knows what’s going to be discussed in each section. You want people to actually read the article, so make it as easy as possible for them to do so.

Cut down on the quotes. Your aim is to summarise the various arguments for the reader, in simple language. A quotation is useful only when if the exact wording in someone else’s piece is critical to your argument. Never more than 2-3 in a piece.

Smaller points

On the issue of the distinction between ME and CFS, the ms opens by describing these conditions as interchangeable, “Myalgic Encephalomyelitis (ME), also known as Chronic Fatigue Syndrome (CFS) or ME/CFS…”.  But then later, this assumption is questioned. One way to reconcile these apparent discrepancies would be to focus almost entirely on the condition referred to in the literature as CFS, and note only briefly that this condition is often referred to by the alternative name ME.  Then if you wanted to, you could include a footnote to say that not all researchers consider the terms ME and CFS to be completely interchangeable; some definitions of ME describe a condition that is more narrowly-defined than CFS (then give an example). Once you’ve dealt with this issue, stick to one term throughout.

Lines 595-598, p. 15. “The researchers note that…”. Omit this whole paragraph. Arguing about whether blinding is impossible or merely impractical doesn’t add anything to your argument.

Author Response

Thank you for reading our article and your comments.

We have shortened the article, changed the tone so that it is as neutral as possible; made the intro neutral and the article easier to read; reorganised our points based on themes (e.g. issues with patient selection) and cut down on the quotes. We briefly describe the terms ME and CFS and use the latter throughout the article.

We've added a summary of the trial describing the authors’ rationale for conducting the trial, the trial itself, the results, the researchers’ conclusions etc.

We have tightened the supporting evidence and instead of saying what the authors concluded, described what their actual findings were.  

We have tightened the critique of the CFS case definition used in FITNET. We have shortened and changed the part about maternal concern.

We have added a section about strengths and weaknesses of the trial.
We avoid using the term “placebo effect” altogether.

There are people who take the view that belief in one’s improvement is as good as – or the same as – actual improvement.  We discuss why there is a big difference between the two.

As suggested, we have changed unproven boom or bust theory to boom or bust theory.

As suggested we removed the part about the PACE trial and only mention it later on because the NHS trial does. And we have also removed the discussion if blinding is impossible or merely impractical. 

As suggested we let people draw their own conclusions as to why the actometer results were omitted and we have removed the part that refered to our personal experience.
